# The Benefit of Sodium-Glucose Co-Transporter Inhibition in Heart Failure: The Role of the Kidney

**DOI:** 10.3390/ijms231911987

**Published:** 2022-10-09

**Authors:** Edoardo Gronda, Emilio Vanoli, Massimo Iacoviello, Pasquale Caldarola, Domenico Gabrielli, Luigi Tavazzi

**Affiliations:** 1Medicine and Medicine Sub-Specialties Department, Cardio Renal Program, UOC Nephrology, Dialysis and Adult Renal Transplant Program, IRCCS Ca’ Granda Foundation, Ospedale Maggiore Policlinico, 20122 Milano, Italy; 2U.O. Rehabilitative Cardiology Sacra Famiglia Hospital Fatebenefratelli, 22036 Erba, Italy; 3Molecular Medicine Department, University of Pavia, 27100 Pavia, Italy; 4Department of Medical and Surgical Sciences, University of Foggia, 71122 Foggia, Italy; 5UOC Cardiology, Azienda Sanitaria Locale BA, 70132 Bari, Italy; 6UOC Cardiology, Azienda Ospedaliera San Camillo-Forlanini, 00152 Rome, Italy; 7Maria Cecilia Hospital, GVM Care & Research, 48033 Cotignola, Italy

**Keywords:** sodium-glucose co-transporter-2, inhibitors, SGLT2, heart failure, renal failure, glomerular filtration rate, GFR, glomerular hyperfiltration, diabetes

## Abstract

In the essential homeostatic role of kidney, two intrarenal mechanisms are prominent: the glomerulotubular balance driving the process of Na^+^ and water reabsorption in the proximal tubule, and the tubuloglomerular feedback which senses the Na^+^ concentration in the filtrate by the juxtaglomerular apparatus to provide negative feedback on the glomerular filtration rate. In essence, the two mechanisms regulate renal oxygen consumption. The renal hyperfiltration driven by increased glomerular filtration pressure and by glucose diuresis can affect renal O_2_ consumption that unleashes detrimental sympathetic activation. The sodium-glucose co-transporters inhibitors (SGLTi) can rebalance the reabsorption of Na^+^ coupled with glucose and can restore renal O_2_ demand, diminishing neuroendocrine activation. Large randomized controlled studies performed in diabetic subjects, in heart failure, and in populations with chronic kidney disease with and without diabetes, concordantly address effective action on heart failure exacerbations and renal adverse outcomes.

## 1. Introduction

Heart failure (HF) studies are mostly designed to prove the efficacy of therapeutic interventions in patients with a predefined cut-off value of left ventricular ejection fraction (LVEF) as primary entry criteria. On the other hand, a predefined low value of the glomerular filtration rate (GFR) may be applied as an exclusion criterion. Thus, on one side, this decision-making process recognizes HF as a “cardiorenal condition”, while on the other side, investigation of HF therapies’ aim to establish their benefit on cardiac outcome, while renal outcomes are often not rated as a concurrent target.

Recently, the gliflozins, inhibitors of sodium-glucose co-transporter 2 (SGLT2i), were cleaved in the cardiovascular arena, proving unexpected effectiveness in restraining HF exacerbations, while they were investigated as hypoglycemic drugs in type 2 diabetes mellitus patients (T2DM) [1]. The pharmacologic action of this class of drug is to inhibit the reabsorption of Na^+^ and glucose in the proximal segment of the glomerular tubule; therefore, they work with exquisite renal effect. In the following studies, the extension of benefit of those molecules on HF outcomes was coupled with concurrent decrease in renal adverse outcomes in a fashion that made clear the need to overcome the heterogeneity in the reporting of kidney function, kidney outcomes, and definitions for kidney end-points in clinical trials. This step is now mandatory in order to gauge the benefit of interventions across large-scale studies performed in T2DM, chronic kidney disease (CKD) and HF [2].

This recent appraisal on the relevance of kidney end-points’ relevance suggests the kidney physiology deserves appropriate consideration when cardiologists analyze investigational data generated by new therapeutic interventions. Accordingly, we aim to analyze how clinical conditions affecting kidney function entail the derangement of renal oxygen consumption, unleashing a detrimental neuro-hormonal overactivation with a harsh prognosis impact on total and cardiovascular mortality.

## 2. Body Fluid Homeostasis Is Centered on Kidney Function

The renal fluid and electrolytes excretion allow the adaptation of body fluid and electrolytes content to the physiological needs. The body fluid balance is intrinsically connected with Na^+^ Cl^−^ K^+^ homeostasis that is, in turn, locally involved in preserving kidney vitality and function. 

The kidney represents only 0.5% of total body weight receiving, albeit with ~25% of cardiac output [3], while its physiologic function needs to extract only 10–15% of available oxygen supply (corresponding to the 7% of the whole O_2_ body consumption) [4]. Thus, a huge disparity clearly exists between renal mass, renal O_2_ consumption and, more so, the amount of delivered renal O_2_.

Indeed, only a minor portion of the very high amount of arterial blood delivered to the kidney contributes to the glomerular filtrate generation. In an animal model, the renal fractional extraction of O_2_ declines when renal blood flow increases, while renal parenchymal O_2_ remains unchanged [5], unveiling the function of the unique arrangement of the preglomerular vessels architecture. This specific arrangement presides over the diffusional shunting from arteries to veins, restricting the blood flow directed toward the glomerular vasculature. This mechanism contributes to the dynamic regulation of intrarenal oxygenation and directly affects the GFR generation with proportional increase in the filtered Na^+^.

A further step in regulation of the GFR generation takes place in the glomerular vasculature that provides hydraulic regulation of pressure in and out of the glomerulus, and that is prominently based on neural endocrine response [6].

Due to the complex renal hemodynamic regime, an intrarenal metabolic control with specific effects on the vasculature and transport systems of cortex and medulla is at the base of the kidney function [4,6]. Many small molecules with short action duration, generated by local demand (nitric oxide, bradykinin, endothelin, angiotensin II, and prostacyclin), concur with local signaling cascades regulating renal function [6]. 

In the kidney, the largest amount of O_2_ consumption occurs in the cortex, where the blood flow is prominently directed to generate the GFR that, in turn, is linked to the Na^+^ reuptake in the proximal tubular segment. The reabsorption process is based on ATP utilization. In the normal subject the average GFR is 180 L/day while the normal urine output ranges from 1.2 to 1.8 L/day, meaning that roughly 1% of the glomerular filtered load is excreted [7] (Figure 1).

Normal daily glomerular filtrate contains ~1 mol (~180 g) of glucose [3] that, if eliminated into the urine, would generate an energy cost equivalent to ~1/3 of the body’s total caloric expenditure. Conversely, in the S1 segment of the glomerular proximal tubule the type 2 sodium-glucose co-transporter (SGLT) 2 reabsorbs 80–90% of the filtered glucose, coupled one to one with Na^+^. The remaining 10–20% of glucose is reabsorbed by SGLT1 in the S2/S3 segment by coupling one sugar molecule with two Na^+^ atoms. Thereby, SGLT1 uses as much as twice the energy per one reabsorbed glucose molecule as SGLT2. The synergy of both sodium-glucose co-transporters accounts for a maximum reabsorption capacity of ~2.5 mol (450 g) of filtered glucose per day [7].

As SGLT1’s reuptake of glucose requires double the Na^+^ as SGLT2, the higher maximum tubular transport of glucose not only entails higher exposure to hyperglycemia but also disposes to fluid retention [8].

The activation of all those sophisticated mechanisms linked to sodium and glucose reabsorption highlights the amount of energy consumed by the selective regulation of solutes excretion and by the reabsorption of Na^+^ and water from the glomerular filtrate. 

This energy consuming process makes the GFR production the major determinant of kidney O_2_ consumption, strongly connected to the entire renal metabolism [6,8,9].

This high-energy process leaves less oxygen available to the medullar vasculature [3,6]. As a consequence, the GFR, with the connected tubular Na^+^ load, affects the O_2_ available for transport work in the cortex and, more critically, in the medulla.

These processes must be tightly coordinated to avoid fluid and electrolyte losses and the intrarenal mechanisms driving the punctual adjustment of the single glomerulus function, include glomerulotubular balance and tubule-glomerular feedback (TGF) [6].

Any increase in the reabsorption process, accounting for the glomerulotubular balance mechanism, is linked to the plasma oncotic pressure resulting after filtration, driven by the intraglomerular pressure gradient. The glomerulotubular balance mechanism takes place almost entirely in the proximal tubule and in the cortical thick ascending limb of Henle, locally involving the highest O_2_ demand driven by the sodium–potassium ATPase, which provides sodium potential to reabsorb solutes from the filtrate [4,6] (Figure 2).

It is important to note that the SGLT1 protein is also present in the small intestine, where it acts as rate-limiting factor for absorption of glucose and galactose, by using the transmembrane sodium gradients to drive the cellular uptake of these molecules. In the multiethnic population of the Atherosclerosis Risk in Communities (ARIC) study, it was observed that a specific modification of the SGLT1 haplotype, qualified by the sequence N51S/A411T/H615Q, was associated with protection from postprandial hyper-glycemia. Those data support the hypothesis the SGLT1 inhibition may result in the decline of T2DM, HF, and mortality incidence by restraining the postprandial glucose levels in subjects at risk [10].

## 3. The Adenosine Hypothesis

Among the reabsorbed solutes, glucose has a prominent role in determining detectable changes in Na^+^ concentration in the glomerular filtrate. This depends on the fact other solutes such as amino acids, phosphates, etc. have a relatively constant plasma concentration, while plasma glucose concentration variably changes in relation to fasting or meals or because of stress-related higher glycemia, accounting for significant variation in glycemic status. Those changes entail variable glucose concentration in glomerular filtrate imposing wavering reabsorption of Na^+^ glucose by SGLT1/2, affecting changes in the Na^+^ concentration delivered to juxtaglomerular apparatus (JGA).

The JGA senses Na^+^ concentration by reabsorbing the atom through the electrochemical sodium potential generated by the basolateral Na^+^/K^+^-ATPase breaking adenosine triphosphate (ATP) down into adenosine diphosphate plus adenosine. Locally, adenosine acts as a vasoconstrictor on adjacent afferent arteriole [6], and Na^+^ fluctuations sensed by JGA influence the inverse relationship between the tubular Na^+^ concentration and the nephron GFR. This minute-to-minute regulation of glomerular flow has direct consequence on glomerular filtration and, ultimately, on fluid and solute reuptake, which correlates to renal O_2_ consumption (Figure 3).

It is also important to recall that adenosine not only balances glomerular oxygen consumption, but also provides contemporary inhibition of renin release [12,13]. The overall investigational data support the action of this autacoid as a prominent contributor to the renal and cardiovascular protection benefits observed in sodium-glucose co-transporter inhibitors clinical trials.

## 4. Sodium-Glucose Co-Transporters Over-Expression: The Related Consequences

In individuals with diabetes mellitus, as Na^+^ and glucose are co-transported, the enhanced reabsorption of glucose is coupled with higher body sodium content, and not surprisingly, 60 to 70% of patients with T2DM may become hypertensive [14]. Consistently in persons with diabetes, the increased exposure of proximal tubular cells to filtered glucose promotes SGLT2 overexpression with further enhancement of glucose reabsorption, leading to a paradoxical increase in the urinary glucose excretion threshold [7,15]. Moreover, the persistence of an uncontrolled glycemic ambient induces the increase in SGLT1/2 mRNA expression by 36% and 20%, respectively, leading to further Na^+^ load reabsorption. In this circumstance, SGLT1/2 activity may account for as much as more than 14% of renal Na^+^ reabsorption along the whole tubule [15], with a significant dip in Na^+^ load reaching the JGA. The decreased amount of Na^+^ passing through the JGA not only reduces local adenosine production, but also attenuates the control on renin angiotensin II axis. In turn, AT II not only restrains the efferent arteriole section, but also reduces the glomerular capillary surface area, impacting hydraulic glomerular conductivity with convergent mechanisms in increasing glomerular filtration pressure [16,17] (Figure 3). 

In general, the accepted marker of high intraglomerular pressure is the glomerular hyperfiltration defined as a GFR ≥ 135 mL/min·1.73 m^2^: beyond this value, glomerular hyperfiltration hurts the mesangial structure, allowing albumin leakage in the filtrate and paving the way toward end-stage diabetic nephropathy [15,16,17].

In addition to these adverse actions, angiotensin II enhances glycemia together with SGLT2 expression in proximal tubular cells, contributing to the metabolic derangement [17]. In the long run, the intraglomerular hemodynamic changes cause profound glomerular damage, leading to progressive CKD and deeply impairing the systemic hemodynamic balance. Indeed, the higher Na^+^ proximal tubule reabsorption draws further water, adding to passive sodium reabsorption that aggravates fluid retention and tissue congestion [8]. The mechanisms concur to explain why local kidney regulatory action may become a critical aspect of renal perfusion in the HF syndrome independently by coexistent diabetes. Low cardiac output results in arterial vasculature under-filling, causing arterial blood to be shunted from the kidneys to the systemic circulation. The disproportionate decrease in renal fraction, secondary to cardiac output, critically enhances the renal efferent sympathetic activity [18]. In this condition, norepinephrine spillover strikingly increases in the kidney, becoming a prominent index of unfavorable outcome independent of left ventricular performance, GFR and overall sympathetic activation [11]. Norepinephrine has a potent action in enhancing peripheral vascular resistances restricting end-organs perfusion while it enhances the glycemic level, leading to increased SGLT2 expression in the nephron and insulin secretion. In two cohort studies performed in patients hospitalized for acute decompensated HF, the blood glucose levels were linked to worse outcomes, irrespective of the diabetes status. In the first study, at a median follow-up of 1.8 years, a greater risk of mortality was present in subjects with higher glucose level at hospital admission in comparison to patients without either blood glucose elevation or diabetes [19]. In a second cohort study, the occurring glycemic variability, but not mean hospital glucose level, was associated with inpatient mortality [20]. Pathophysiology and investigational data link neurohormonal changes and failing glycemic control to HF progression, whether or not diabetes is diagnosed.

## 5. Impact of SGLT2 Inhibition on Renal Function

As the sodium-glucose co-transporters play a crucial role in renal function and in cardiovascular balance, it is not surprising that the administration of SGLT2is widely affects kidney, heart and vasculature physiology, endowing clinical benefits that may appear oversized in respect to the known mechanism of action. 

In fact, the use of SGLT2 inhibitors in the presence of deranged renal hemodynamic conditions aims to restore the filtrate flow to the thick ascending limb and the Na^+^ delivery to the juxtaglomerular apparatus, thereby increasing local adenosine generation and re-establishing afferent arteriole tone via tubuloglomerular feedback. The net effect of their pharmacologic action is translated in decreasing the glomerular filtration pressure [7,8,15] (Figure 3).

The administration of SGLTIs in T2DM, as well as in HF and CKD conditions, independently of diabetes presence, drives acute eGFR changes, characterized by dose-dependent reductions of ≈5 mL/min·1.73 m^2^ over several weeks [7,8]. Subsequently the eGFR tends to return toward baseline and remains stable over time. In experimental conditions, the dip reverses after treatment of 3 to 4 weeks or within 2 weeks of drug discontinuation, indicating the functional nature of the changes [7,8,15]. As a matter of fact, in the long-term follow-up of cardiovascular outcome trials, the GFR decline was progressively more severe in the control arm, independently of the investigated clinical condition, suggesting that SGLT2 inhibition can be effective in preserving kidney function. It has to be highlighted that data from the DAPA HF study post hoc analysis depicted the eGFR dip following dapagliflozin administration. The dip was modest and present either in patient with and without T2DM, but diabetics experienced a mildly higher dip [21]. A higher eGFR dip has been coupled with a better outcome, particularly in older subjects with a lower eGFR and with a higher ejection fraction [22]. 

Intriguingly, in the DAPA-CKD trial, dapagliflozin protected the kidneys of patients with CKD, regardless of the presence or absence of T2DM [23]. The point marks the relevant interest as the glucosuric effect of SGLT2 inhibitors is linked to the glycemic status and to the filtered glucose, as well as to the drug dose effectiveness. In the study, the SGLT2 inhibition proved kidney-protective effects also in non-diabetic patients with more severe CKD who filter less glucose and, thus, show little glucosuria or effects on blood glucose levels. The reason for this unexpected benefit may reside in the fact that a small decline in glucose and in Na^+^ reabsorption in the proximal tubule is followed by large decline in glomerular filtration pressure [24]. 

The SGLT2is actions on renal function are both important mechanistically and clinically, because the underlying changes in intraglomerular flow and pressure are likely to play an effective role in driving the observed clinical benefits, especially when linked to renin angiotensin aldosterone system (RAAS)-blocking agents. The RAAS inhibitors decrease intraglomerular pressure through efferent arteriolar vasodilatation, resulting in nephron preservation. In the RENAAL study, the administration of losartan in patients with T2DM was followed by GFR dip. Those patients who experienced the greatest tertile of GFR decline within 3 months of the initiation of therapy later enjoyed the best preservation of renal function [25]. 

In the EMPA-REG OUTCOME trial exploratory analysis, the short-term and long-term effects of empagliflozin on albuminuria in patients with type 2 diabetes and established cardiovascular disease documented a rapid and sustained reduction in albuminuria according to patient baseline albuminuria status. Such an effect was reasonably mediated by the small early decrease in GFR. Thus, the albuminuria status decline can probably be used as a short-term marker of intraglomerular pressure and renal efficacy for these types of drugs [26].

By analyzing data of the SGLT2 inhibitors large-scale studies investigating diabetes, HF and CKD, the clinical effects induced by their pharmacological action can be split in three consecutive steps: the first involves the direct hypoglycemic action that is soon followed by the restraining of HF exacerbations and later by the decline in renal adverse outcomes.

## 6. Impact of SGLT2 Inhibition on Heart Failure Progression

The SGLT inhibitors displayed immediate pharmacologic action on glycemic control. However, hyperglycemia is a weak risk factor for the development of cardiovascular disease [27], and the SGLT2i glycosuric effect reduces the HbA1c by 0.8–1.0%, providing a greater glycemic reduction in subjects with higher HbA1c [7,15]. Nevertheless, by reducing the maximum tubular transport of glucose and the glycosuria threshold, SGLT2 inhibitors enhance glucose excretion, leading to the reduction in fasting and postprandial plasma glucose levels and the improvement in insulin secretion and insulin sensitivity. However, when renal function is critically reduced, the SGLT2is hypoglycemic action is also reduced, requiring further adjustment of diabetes therapy [7,15].

A specific setting of SGLT inhibitors pertains to the immediate functional fall in intraglomerular filtration pressure that follows the inhibition of Na^+^ glucose co-transporter. Without exceptions, the earliest clinical benefit obtained by SGLT2i administration has been the decline in HF hospitalizations in diabetic, HF and CKD patients (Appendix A). SGLT2 inhibition was consistently coupled to a rapid decline in HF exacerbations, irrespective of the study population. Importantly, in all cardiovascular outcome trials, the outcome actuarial curves maintain stable diverging trajectories along the entire follow-up for either HF patients with reduced left ventricular ejection fraction (HFrEF) or in those with preserved ejection fraction (HFpEF). It has to be noted that in DAPA HF [28], in EMPEROR-Reduced [29], and in EMPEROR-Preserved [30] studies, the occurrence of HF hospitalization and of cardiovascular death decline started to be statistically significant, while GFR dip was observed after SGLT2 inhibition. Concordantly, the authors of DAPA-HF, after the start of dapagliflozin, observed the early dip in GFR was associated with better clinical outcomes than GFR decline observed on the placebo arm [22], thus highlighting the role of filtration pressure decline in HF prevention, independently of left ventricular ejection fraction (Figure 4, Appendix A), as was recently recognized [31]. In both studies, the limited decline of NT pro-BNP plasma concentration (−24% after dapagliflozin and −5% after empagliflozin) [28,29] was below the threshold, addressing a possible clinical benefit [32] supporting that the drug effect was not primarily in the heart.

Data from a post hoc analysis performed on the DAPA CKD study population compared patients with and without HF at baseline, further supports this concept. The GFR dip after SGLT2 inhibition was comparable in the two study subgroups, but a more prominent HF hospitalization (HFh) rate decline was present among subjects with HF at baseline (HR 0.51; 95% CI: 0.34–0.76 vs. HR 0.62; 95% CI: 0.35–1.10), while overall mortality was consistently affected in both subgroups (HR 0.56; 95% CI: 0.34–0.93 vs. HR 0.73; 95% CI: 0.54–0.97) [33] (Appendix A). In both HFrEF studies, the strict relation between the dip in glomerular filtration pressure and the decline in HF exacerbations is strengthened by the consistency of the better outcome achieved by patients receiving an angiotensin receptor neprilysin inhibitor (ARNI) in comparison to those who did not. In the EMPEROR-Reduced study design [29], the effect of the association between ARNI and empagliflozin was a pre-specified end-point, and the statistical analysis ruled out interaction between the two drugs. It must be recalled that neprilysin, the inhibitor of natriuretic peptide, is prominently allocated in the tubular structure of the glomerulus. The data highlight that the result has been achieved by summing up the SGT2i benefit with the ARNI one. Both drugs lead, in synergy, to a reduction in the glomerular filtration pressure, as SGLT2is increase the afferent arteriole tone and ARNI decreases the tone of efferent arteriole, engaging the glomerulotubular balance mechanism. Coherent with this, in the EMPEROR-Reduced post hoc analysis, the association of sacubitril–valsartan to empagliflozin drew a larger decline in renal adverse outcomes in comparison to the patient subgroup treated with ACE inhibitors/Angiotensin II receptor blockers [34], although the event reduction was not statistically significant, due to the limited size of the subgroups.

The impressive power of SGLT inhibition in preventing HF exacerbation is also highlighted by two sotagliflozin studies that were prematurely halted due to loss of sponsor funding. Notably, the molecule provides contemporary inhibition of SGLT2 and SGLT1.

In the SOLOIST-WHF study, sotagliflozin was investigated in patients with diabetes and recently worsening HF [35], while in the SCORED study, the molecule was investigated in subjects with diabetes and CKD [36]. Despite the shortened follow-up, in both investigations, the drug administration resulted in a significantly lower total number of hospitalizations and urgent visits for HF than placebo therapy (Appendix A).

In one other large-scale cardiovascular outcome study performed with ertugliflozin, SGLT2 inhibition was investigated in patients with T2DM and established atherosclerotic cardiovascular disease. While the study’s primary outcome for superiority, based on three points MACE, was missed (HR 0.97; 95% CI: 0.85–1.11) [37], the drug proved to provide an early and sustained decrease in the risk of first HFh (HR 0.70; 95% CI: 0.54–0.90) (Appendix A), and previous HFh did not modify this effect (HR 0.63; 95% CI: 0.44–0.90] [38].

The overall concordance of cardiovascular outcome trials suggests that SGLT2is can restrain HF progression in the global cardiovascular population on top of the contemporary GDMT, independently of the left LVEF value.

## 7. Impact of SGLT2 Inhibition on Renal Adverse Outcomes

While the SGLT2i action lowers glomerular filtration pressure affecting HFh, it also adds further clinical benefit by preserving the glomerulus integrity in the kidney. In the SGLT2i controlled studies-treated arm, the initial GFR dip was followed by partial recovery and relative stabilization. In the control arm, an observed decrease in GFR steadily worsened, crossing the treated arm line with increasing adverse outcomes. In the studies performed on clinical conditions in patients with diabetes with different cardiovascular risk profiles (EMPA-REG Outcome [39], CANVAS [40], DECLARE–TIMI 58 [41]), with advanced HFrEF (EMPEROR-Reduced [29]) and with CKD, associated or not with diabetes (CREDENCE [42], DAPA CKD [43]), the better-preserved renal function translates to a statistically significant lower occurrence in renal adverse outcomes. Thus, again, the early benefit on HF progression in cardiovascular outcome trials was followed later by a decrease in adverse renal outcomes, consistent with the constant relation between the preservation of kidney function and the restraining of HF adverse events (Appendix A).

In several studies, the impact of SGLT2 inhibition on the burden of adverse events was large enough to affect cardiovascular mortality and all-cause mortality (Appendix A) (EMPAREG outcome [39], DAPA HF [28], CREDENCE [42], DAPA CKD [43]) to a degree that had not been observed since ACE-inhibition was introduced [8]. In the CREDENCE study, mortality was not quite statistically significant (HR 0.83; 95% CI 0.68–1.02). The reason may reside in the severe prognosis of enrolled patients that were diabetics with albuminuric CKD, a condition that portends advanced disease with an associated high mortality—twice as high compared to the control arm of DAPA CKD. Nevertheless, the correlation between the decline in renal-specific composite outcome and of death from any cause was also evident in the CREDENCE Study results (Appendix A). 

It is worth noting that in the DAPA CKD study, dapagliflozin’s benefit on HF occurred shortly after randomization, while in the treated arm, the incidence of renal-specific composite outcomes started to diverge from the control arm at approximately 6 months after randomization, and it became statistically significant after 18 months (HR 0.61; 95% CI: 0.51–0.72) (Figure 4, Appendix A) [43], contemporary to the decline in death from any cause (HR 0.69; 95% CI: 0.53–0.88) (Appendix A) [44]. 

In the EMPEROR-Reduced study, the renal adverse outcomes dip reached statistical significance in the empagliflozin-treated arm (HR 0.50; 95% CI 0.32–0.77). Of note, the actuarial curves started to diverge approximately after 220 days from randomization and reached the widest divergence after 720 days (11 months) when the GFR decline in the control arm was continuing, but not overlapping to the treated arm value [45]. 

As previously addressed among large-scale trials conducted in HF, cardiovascular mortality was significantly affected only in the DAPA HF study (Appendix A). In the study, the benefit size included a reduction in all-cause mortality (HR 0.83; 95% CI 0.71–0.97). Importantly, the survival curves did not start to diverge in concomitance to the HF hospitalization rate decline, but only after ten months of treatment with the contemporary occurrence of a trend toward a decline in renal adverse outcomes [46].

Furthermore, in the treated arm of cardiovascular outcome trials, the SGLT2i administration restrained the renal adverse outcomes incidence after the GFR dip was partially recovered, achieving relative stability [42,43,44,45,46]. The better patient outcome provided by the inhibition of sodium and glucose reuptake has been consistent with the decrease in filtration pressure coupled with sparing of nephron oxygen consumption and of renal degradation. In the control arm of large-scale cardiovascular investigations, the initial GFR dip did not occur, but the slow and progressive GFR decline took place throughout the entire investigation, coupled with the higher incidence of renal adverse outcomes. 

Further confirmatory evidence comes from the EMPA-HEART (Effects of Empagliflozin on Cardiac Structure in Patients With Type 2 Diabetes). In the sub-study investigation conducted in diabetic subjects with chronic coronary artery disease, the empagliflozin administration restored erythropoietin levels, driving higher hematocrit and lower ferritin and red blood cell hemoglobin concentrations, after the six-month follow-up. The authors suggest that the restoration of erythropoietin production was due to the decreased renal O_2_ [9]. The persistence of higher hematocrit after SGLT2is administration proved to favorably impact patient outcomes in investigated diabetic and HF populations [21,47].

Finally, besides the results of EMPEROR-Preserved, those of the DELIVER trial have been recently presented. Analogously to empaglifozin, dapaglifozin was able to reduce the events related to heart failure progression in patients with LVEF > 40% [48]. However, the data about the effects on renal function have not yet been presented.

## 8. Conclusions

In the large controlled studies performed with SGLT inhibitors, there is a consistent decline in cardiovascular outcomes across the different investigated clinical conditions. The overall action of gliflozins on metabolism, cardiac and renal physiology results in three key clinical outcomes: better glycemic control, a decline in HF exacerbations, and a slowing of renal function deterioration by sparing glomerulus function, thus aligning the drug benefit with the improved overall life expectancy.

The recent investigational data generated by large controlled studies conducted with SGLT2is in HFrEF and HFpEF as well as in T2DM and CKD confer strength to the hypothesis that the drugs’ benefit begins in the kidney. 

The whole bulk of data reinforces the concept that in clinical conditions affecting kidney function, for the therapies to be effective, they should decrease the glomerular filtration pressure that does not necessarily mirror the GFR value, which in turn does not represent the efficiency of the organ performance. The concept entails focusing on the organ physiology balance that is expressed by the incidence of adverse outcomes, mostly cardiac and renal in the SGLT inhibitors case, and not on a function index to estimate a specific therapeutic benefit.

## Figures and Tables

**Figure 1 ijms-23-11987-f001:**
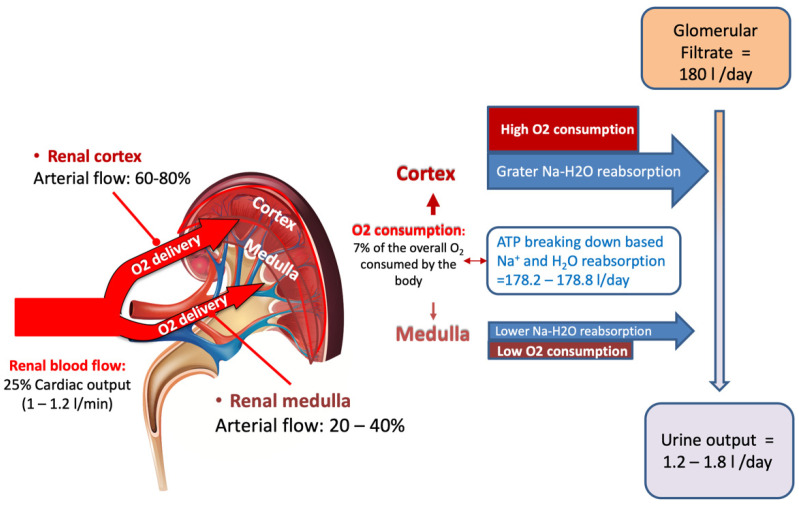
Despite renal mass accounts only for 0.5% of total body mass, 25% of cardiac output is delivered to the kidney, which consumes the 7% of the whole body oxygen consumption. Renal oxygen consumption is mostly connected to the selective reabsorption of glomerular filtrate that corresponds to approximately to 180 L/day, while daily urine output is restricted to 1.2–1.8 L/day. The largest reabsorption work takes place in the proximal segment of glomerular tubule in the cortex area, where arterial blood flow provides the highest oxygen delivery to supply energy for urine concentration and selective solutes excretion (see text for details).

**Figure 2 ijms-23-11987-f002:**
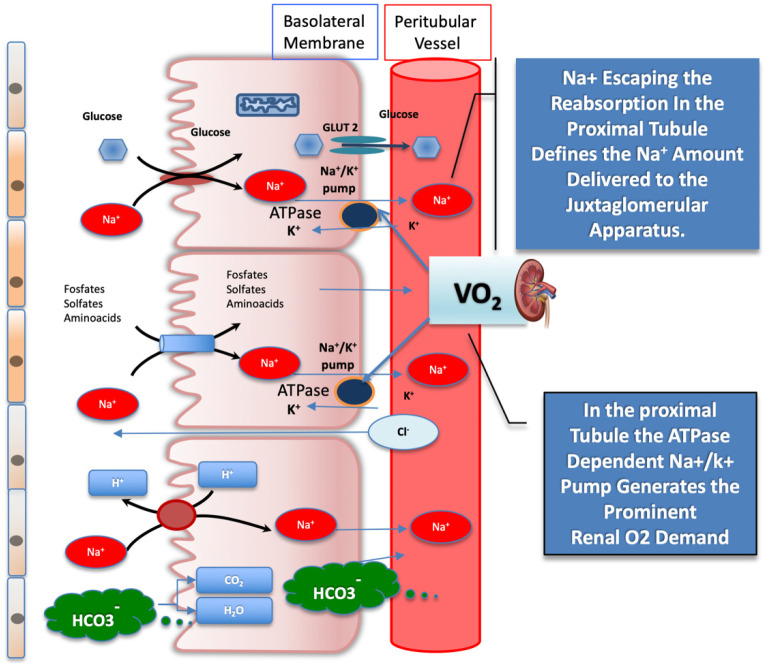
Schematic illustration of filtrates reabsorbed from the glomerular proximal tubule activated by sodium-glucose co-transporter (SGLT) 2, which operates in the glomerular proximal tubule. The SGLT2, the bicarbonate reabsorption, and the Na^+^/K^+^ ATPase, which provides sodium energy to drive both, are represented. The active reuptake of bicarbonate and of other solutes, coupled with Na^+^, drives the highest tissue O_2_ consumption in the kidney. The amount of Na^+^ escaping the reabsorption in the proximal section tubule is sensed by the juxtaglomerular apparatus and sets the afferent arteriole tone. See text for details. GLUT2 = glucose transporter 2; SGLT2 = sodium-glucose co-transporter 2.

**Figure 3 ijms-23-11987-f003:**
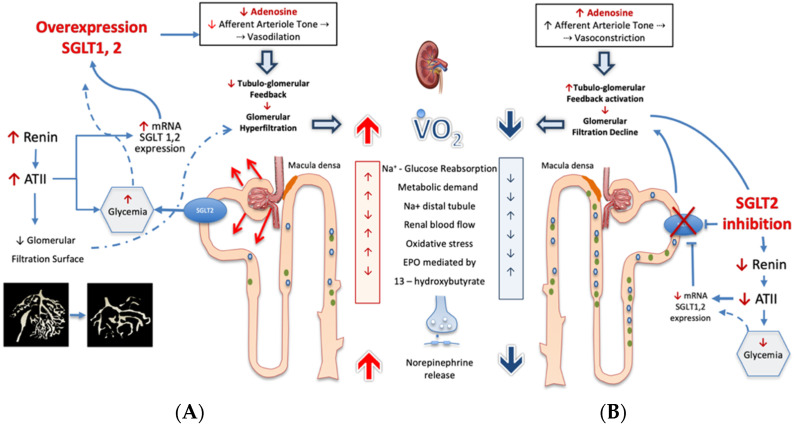
Under conditions of ambient hyperglycemia (panel (**A**)), sodium-glucose co-transporter 2 activity (SGLT2) is increased, thereby reducing the juxtaglomerular apparatus delivery of Na^+^. This affects the tubuloglomerular feedback mechanisms, as the concentration of Na^+^ that transits across the juxtaglomerular apparatus establishes the adenosine triphosphate (ATP) release and its breakdown to adenosine. Local adenosine concentration acts by vaso-constricting afferent arteriole. Thus, any adenosine concentration decrease may impact the appropriate regulation of glomerular blood flow and of glomerular filtration pressure and, therefore, of renal metabolic work. An unfavorable consequence of the higher metabolic work is the increased cortical oxidative stress-induced tubulointerstitial damage that causes a fall in erythropoietin production [9]. The adenosine generation in the juxtaglomerular apparatus is linked to a large network of biological effects such as the control of renin release; therefore, a decreased breakdown of ATP in the juxtaglomerular apparatus may lead to enhanced ATII generation. Angiotensin II not only affects the circulation balance systemically, but also compromises the glomerulotubular balance by restraining the efferent arteriole section and the glomerular capillary surface area. All together, these responses account for the incremental neural transmitter release, leading to the exceedingly high renal norepinephrine spillover occurring in heart failure progression [11]. Under these conditions, the SGLT2 inhibition (**B**), by inducing glucoresis and natriuresis, is able to re-establish the distal tubular flow and Na^+^ delivery to the juxtaglomerular apparatus. In this way, SGLT2 inhibition restores local adenosine generation that reestablishes the afferent arteriole tone, leading to lower filtration pressure and to lower metabolic work while it restores control on neuroendocrine pathway. See text for details. SGLT2 = sodium-glucose co-transporter-2; VO2 = Oxygen consumption; ATII = angiotensin II; EPO = erythropoietin.

**Figure 4 ijms-23-11987-f004:**
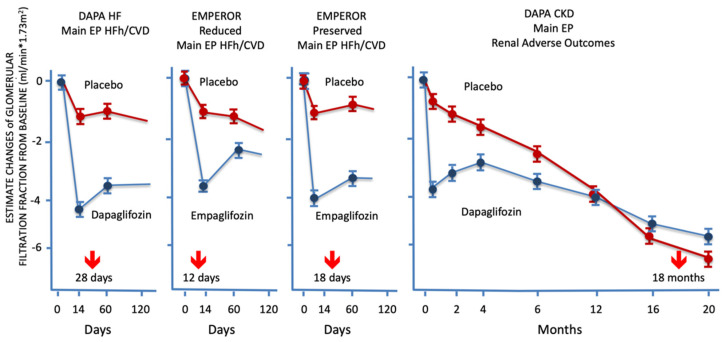
The figure depicts the temporal relationship between the detection of the statistically significant achievement of the primary end-point of DAPA HF [28], EMPEROR-Reduced [29], EMPEROR-Preserved [30], and DAPA CKD [22] studies and the glomerular filtration rate (GFR) dip occurrence after inhibition of sodium-glucose co-transporter-2. The significant decrease in the combined end-point of heart failure (HF) hospitalization or cardiovascular death was achieved in HF studies while the GFR dip was occurring, in a time window between two and four weeks from randomization. In those studies, the decline in HF exacerbations largely drove the therapeutic benefit. In the DAPA CKD, the decline in the primary end-point of renal adverse outcomes (RAO) started to appear after 6 months in the study follow-up, but statistical significance was achieved at 18-month follow-up. Of note, in all SGLT2is investigations, the decline in RAO was detected at a significant distance from the occurrence of GFR dip. See text for details. SGLT2 = sodium-glucose co-transporter-2; GFR = glomerular filtration rate; EP = end-point; HFh = heart failure hospitalization; CVD = cardiovascular death.

## Data Availability

Not applicable.

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
