# Peer review of "The Benefit of Sodium-Glucose Co-Transporter Inhibition in Heart Failure: The Role of the Kidney"

_ijms, 2022, doi:10.3390/ijms231911987_

Round 1

Reviewer 1 Report

There are many spelling and misuse examples.

The authors fail to consider the overlap of SGLT 2 in kidney and SGLT 1 in bowel.

A number of proposed mechanisms proposed for SGLT 2 inhibitors are speculative.

This paper adds little to the comprehensive review by Heerspink NEJM 2019.

Author Response

We would like to thank the reviewer for his/her helpful comments. This is our point to point reply:

  • There are many spelling and misuse examples.

Response:

We have checked the text and corrected when found.

  • The authors fail to consider the overlap of SGLT 2 in kidney and SGLT 1 in bowel.

Response:

We thank the reviewer or his comment, we added at line 144 the following sentence:

“On note the SGLT1 protein is also present in the small intestine where it acts as rate-limiting factor for absorption of glucose and galactose, by using the transmembrane sodium gradients to drive the cellular uptake of these molecules. In the multiethnic population of the ARIC (Atherosclerosis Risk in Communities) study it was observed that a specific modification of the SGLT1 haplotype, qualified by the sequence N51S/A411T/H615Q, was associated with protection from postprandial hyper-glycemia. Those data support the hypothesis the SGLT1 inhibition may result in the decline of T2DM, HF, and mortality incidence by restraining the postprandial glucose levels in subjects at risk.”

And the relative reference:

  1. Seidelmann SB, Feofanova E, Yu B, Franceschini N, Claggett B, Kuokkanen M, Puolijoki H, Ebeling T, Perola M, Salomaa V, Shah A, Coresh J, Selvin E, MacRae CA, Cheng S, Boerwinkle E, Solomon SD. Genetic Variants in SGLT1, Glucose Tolerance, and Cardiometabolic Risk. J Am Coll Cardiol. 2018;72:1763-1773. DOI: 10.1016/j.jacc.2018.07.061

  • A number of proposed mechanisms proposed for SGLT 2 inhibitors are speculative.

Response:

We agree with the reviewer. Most of the evidence about the mechanisms underlying the favorable effects of SGLT2i are speculative but we think that the current evidence strengthen the mechanisms we have focused on.

  • This paper adds little to the comprehensive review by Heerspink NEJM 2019.

Response:

We are really sorry about the point raised by the reviewer. We disagree about it. Firstly, we tried to offer a point of view about the mechanisms leading to the favorable effects of SGLT2i which is not reported in most of the reviews focused on this issue.

Secondly, at the best of our knowledge as well as based on research in pubmed and scopus, there is no review on NEJM in the 2019 with Heerpsink as first autor.

The review published by this author in 2018 (Heerspink HJL, Kosiborod M, Inzucchi SE, Cherney DZI. Renoprotective effects of sodium-glucose cotransporter-2 inhibitors. Kidney Int. 2018 Jul;94(1):26-39.) reports only some of the points that we have focused on.

Reviewer 2 Report

I feel, I should congratulate you on the idea to review the complexed pathophysiology of heart - kideny interaction in heart failure, especially in the context of SGLT2 inhibition. Very good job! 

Author Response

We would like to thank the reviewer for having appreciated our paper.

Reviewer 3 Report

This is a very useful and organized review paper and the figures are clear and beautiful. Please change the title to one sentence. 

Author Response

This is a very useful and organized review paper and the figures are clear and beautiful. Please change the title to one sentence. 

Response:

We would like to thank the reviewer for having appreciated our paper. We changed the title into:

The benefit by Sodium Glucose Co-transporter Inhibition in the heart failure: the role of the kidney

Reviewer 4 Report

The role of the kidney in heart failure. The benefit by Sodium Glucose Co-transporter Inhibition by Gronda et al is a very interesting paper. It provides a deeply desciption of SGLT inhibitors from pharmacodynamic to clinical benefit.

I would suggest to add the recent DELIVER-HF trial among the studies reported in the paper (as well in the supplementary table).

Line 213 "The disproportionate decrease of renal fraction of cardiac output critically enhances..." I would suggest "The disproportionate decrease of renal fraction, secondary to cardiac output, critically enhances..."

Author Response

The role of the kidney in heart failure. The benefit by Sodium Glucose Co-transporter Inhibition by Gronda et al is a very interesting paper. It provides a deeply desciption of SGLT inhibitors from pharmacodynamic to clinical benefit.

Response: We would like to thank the reviewer for his/her helpful comments. This is our point to point reply:

  • I would suggest to add the recent DELIVER-HF trial among the studies reported in the paper (as well in the supplementary table).

Response:

It is a very interesting point. However, the published and/or presented results of DELIVER, did not still report the effects of dapaglifozin on renal function in HFmrEF/HFrEF. We reported this aspect at line 421:

“Finally, beside the results of EMPEROR-preserved, those of DELIVER trial have been recently presented. Analogously to empaglifozin, dapaglifozin was able to reduce the events related to heart failure progression in patients with LVEF>40% [46]. However, the data about the effects on renal function have not been yet presented.”

  • Line 213 "The disproportionate decrease of renal fraction of cardiac output critically enhances..." I would suggest "The disproportionate decrease of renal fraction, secondary to cardiac output, critically enhances..."

Response:

Thank you, we corrected it.

Round 2

Reviewer 1 Report

I think the authors make the assumption that the changes in renal hemodynamics typically seen with the onset of hyperglycemia persist as SGLT2 inhibition is maintained. I do not think this is proven. Much of the argument follows what I take to be an unproven assumption.

My suggestion would be clarify the evidence or to make it explicit that these are assumptions that warrant critical evaluation or testing.

Author Response

We would like to thank the reviewer for his/her further comments. This is our reply.

"I think the authors make the assumption that the changes in renal hemodynamics typically seen with the onset of hyperglycemia persist as SGLT2 inhibition is maintained. I do not think this is proven. Much of the argument follows what I take to be an unproven assumption.

My suggestion would be clarify the evidence or to make it explicit that these are assumptions that warrant critical evaluation or testing."

Response:

We tried to better clarify this point at lines 255-272:

"As matter of fact, in long term follow up of cardiovascular outcome trials, the GFR decline was  progressively more severe in the control arm, independent of the investigated clinical condition, suggesting that SGLT2 inhibition can be effective in preserving kidney function. It has to be highlighted that data from the DAPA HF study post-hoc analysis depicted the eGFR dip following dapagliflozin administration, was modest and present either in patient with and without T2DM, but diabetics experienced a mildly higher dip [21]. The higher eGFR dip has been coupled with better outcome particularly in older subjects with lower eGFR and with higher ejection fraction [22].

Intriguingly in the DAPA-CKD trial, dapagliflozin protected the kidneys of patients with CKD, regardless of the presence or absence of T2DM [23]. The point marks relevant interest as the glucosuric effect of SGLT2 inhibitors is linked to the glycemic status and to the filtered glucose as well as to the drug dose effectiveness. In the study the SGLT2 inhibition proved kidney protective effects also in non-diabetic patients with more severe CKD who filter less glucose and, thus, show little glucosuria or effects on blood glucose levels. The reason of this unexpected benefit may reside in the fact that a small decline of glucose and of Na+ reabsorption in the proximal tubule is followed by large decline of glomerular filtration pressure [24]."

We also added the relative references:

21. Petrie, M.C.; Verma, S.; Docherty, K.F.; et al. Effect of Dapagliflozin on Worsening Heart Failure and Cardiovascular Death in Patients With Heart Failure With and Without Diabetes. JAMA 2020, 323, 1353-1368.

22. Adamson, C.; Docherty, K.F.; Heerspink, H.J.; et al. Initial Decline (Dip) in Estimated Glomerular Filtration Rate After Initiation of Dapagliflozin in Patients With Heart Failure and Reduced Ejection Fraction: Insights From DAPA-HF. Circulation 2022, 146, 438-449.

23. Heerspink, H.J.L.; Jongs, N.; Chertow, G.M.; et al.; DAPA-CKD Trial Committees and Investigators. Effect of dapagliflozin on the rate of decline in kidney function in patients with chronic kidney disease with and without type 2 diabetes: a prespecified analysis from the DAPA-CKD trial. Lancet Diabetes Endocrinol 2021, 9, 743-754.

24. Vallon, V.; Thomson, S.C. The tubular hypothesis of nephron filtration and diabetic kidney disease. Nat Rev Nephrol 2020, 16, 317-336.